# Alopecia Areata Occurring after COVID-19 Vaccination: A Single-Center, Cross-Sectional Study

**DOI:** 10.3390/vaccines10091467

**Published:** 2022-09-05

**Authors:** Francesco Tassone, Simone Cappilli, Flaminia Antonelli, Ruggiero Zingarelli, Andrea Chiricozzi, Ketty Peris

**Affiliations:** 1UOC di Dermatologia, Dipartimento di Scienze Mediche e Chirurgiche, Fondazione Policlinico Universitario A. Gemelli-IRCCS, Largo Agostino Gemelli 8, 00168 Rome, Italy; 2Dermatologia, Dipartimento di Medicina e Chirurgia Traslazionale, Università Cattolica del Sacro Cuore, 00168 Rome, Italy; 3Largo A. Gemelli 8, 00168 Rome, Italy

**Keywords:** alopecia areata, COVID-19, SARS-CoV-2, vaccine

## Abstract

Limited data concerning the development of autoimmune skin diseases after COVID-19 vaccination are currently available. Recently, a few reports described the development, worsening or recurrence of alopecia areata after the administration of COVID-19 vaccines. High variability in terms of disease onset following vaccination as well as the heterogeneous topical and/or systemic treatment approaches have been described. **Methods:** All patient-related data and images were obtained as part of clinical routine. Diagnosis of alopecia areata was established according to clinical and trichoscopic findings, along with the exclusion of common differential diagnoses. **Results.** Twenty-four patients, 20 females (83.3%) and four males (16.7%), with a mean age of 39.1 years (age range: 14–66 years), were examined for the occurrence of alopecia areata within 16 weeks after COVID-19 vaccination. Out of 24, 14 patients (58.3%) experienced a patchy alopecia areata, while an extensive disease occurred in 10/24 patients (41.7%): six patients with whole scalp involvement (alopecia areata totalis) and four patients with the whole body affected (alopecia areata universalis). Twelve patients reported a history of autoimmune disease (50%). Treatment with topical corticosteroid was performed in almost all patients with patchy alopecia areata, whilst it was associated with systemic drugs (corticosteroids, minoxidil, cyclosporin) in the case of generalized alopecia areata and alopecia areata universalis. Mean baseline values of Severity of Alopecia Tool (SALT) score decreased from 43.4 to 36.6 after 12 weeks of treatment, with evidence of hair regrowth in 16/21 patients. **Conclusion.** This study described the occurrence of alopecia areata after COVID-19 vaccination and its management that implicates the use of both topical and systemic therapies.

## 1. Introduction

COVID-19 vaccines have been considered potential triggers of a broad spectrum of cutaneous pathologic conditions, including bullous pemphigoid-like, dermal hypersensitivity reactions, herpes zoster, lichen planus-like, neutrophilic dermatosis and urticaria [1,2]. Limited data concerning the development of autoimmune skin diseases after COVID-19 vaccination are currently available [3]. The development, worsening or recurrence of alopecia areata, an autoimmune condition, has been described after COVID-19 vaccination [4,5,6,7,8,9,10]. Alopecia areata is clinically characterized by small patches of baldness localized on the scalp and/or body, with a potential evolution to total scalp hair loss (alopecia areata totalis) and/or total body hair loss (alopecia areata universalis) [4,5,6,7,8,9,10]. The reported time interval between vaccination and the onset of alopecia areata is highly variable, and heterogenous topical and/or systemic therapeutic approaches have been proposed [4,5,6,7,8,9,10]. Scollan et al. described nine cases of alopecia areata that appeared up to 16 weeks after BNT162b2 vaccine and within 4 and 8 weeks with mRNA-1273 vaccine [4]. Further data disclosed the onset of alopecia areata after 1–3 weeks with BNT162b2 vaccine [5,6,9,10]. The exposure to ChAdOx1 nCoV-19 was related to a relatively quick onset of clinical manifestations from a few days to 2 weeks and 3 weeks [6,7,8]. This retrospective observational study aimed to evaluate the broad spectrum of clinical features and therapeutic management of alopecia areata that occurred after COVID-19 vaccination.

## 2. Materials and Methods

A retrospective review of the clinic database dedicated to hair-related disorders at the Department of Dermatology of the Catholic University of Rome, Italy, was undertaken from 31 March 2021–31 March 2022. Using this database-generated list, all charts were reviewed, identifying those patients who were diagnosed with alopecia areata, also following the use of COVID-19 vaccines. For each patient, medical records and photographic documentation were retrieved. The baseline clinical and demographic characteristics (i.e., age, sex, onset of alopecia areata, comorbidities and any concomitant medications) were collected. Vaccine type (BNT162b2, ChAdOx1 nCoV-19, mRNA-1273) and schedule (number of doses received), as well as timing between first or booster dose and appearance of clinical manifestation, were recorded. According to guidelines, diagnosis was established using clinical and trichoscopic criteria, along with exclusion of common differential diagnoses. According to disease extension, alopecia was classified as patchy alopecia areata, alopecia areata totalis and alopecia areata universalis. Data related to Severity of Alopecia Tool (SALT) score values (which range from 0 indicating no hair loss to 100 indicating complete scalp hair loss), as well as details about trichoscopic examination (exclamation mark hairs, black dots and yellow dots), were obtained from patients’ charts as initial visit (baseline visit) and as regularly scheduled follow-up visit (after 12 weeks from baseline). Serum markers for liver, kidney and thyroid function were investigated. Patients referring to the ambulatory dedicated to hair-related disorders within a 1-year period (31 December 2018–31 December 2019) pre-COVID pandemic constituted the control group. Signed informed consent was obtained from patients to extract data from their clinical records. The study was conducted in accordance with the Declaration of Helsinki and upon approval by the Local Ethical Committee—Fondazione Policlinico Universitario Agostino Gemelli IRCCS-Università Cattolica del Sacro Cuore, Prot N.: 4726.

### Statistical Analysis

SPSS Statistics for Windows, Version 25.0 (Armonk, NY, USA, IBM Corp.) was used for the statistical analyses. The mean standard deviation for numerical variables and the number and % values for categorical variables were given as descriptive statistics. The chi-square test was applied to compare percentages of patients affected by alopecia areata, among all hair disorders, between the two groups. *p* value ≤ 0.05 was considered as statistically significant. Percent change in SALT score from baseline to week 12 was assessed.

## 3. Results

Out of 440 charts of patients referring to the ambulatory of hair disorders during the period 31 March 2021–31 March 2022, 74 subjects (16.8%) were affected by alopecia areata. The percentage of patients affected by alopecia areata was in line with the rate of patients (14%, 68/484) with a diagnosis of alopecia areata observed in the same service prior to the COVID-19 pandemic (31 December 2018–31 December 2019). The number of cases occurring during the COVID-19 vaccination campaign was not statistically different from those referring to the same service prior to the COVID-19 pandemic (*p* = 0.244).

The analysis of medical records identified 24 of them (32.4%, 24/74) occurring after COVID-19 vaccination. Twenty females (83.3%) and four males (16.7%), with a mean age of 39.1 years (age range: 14–66 years old) had prior vaccination for COVID-19. Fifteen of 24 (62.5%) patients underwent BNT162b2 vaccination (11 patients received up to the 2nd dose, four patients also the 3rd dose), 5/24 patients (20.8%) were vaccinated with two doses of mRNA-1273 vaccine and 4/24 patients (16.7%) with two doses of ChAdOx1 nCoV-19. No alterations in serum markers for liver, kidney and thyroid function were detected. No recent SARS-CoV-2 infection was recorded. The onset time of alopecia areata was highly variable, ranging from as early as 1 week after vaccination up to 16 weeks after completing the vaccination doses. In detail, alopecia areata occurred 1 to 16 weeks (mean 5.7) after the administration of the BNT162b2 vaccine (12 patients developed alopecia areata after the 2nd dose and three patients after the 1st dose), week 1 to 3 (mean 2.2 weeks) after ChAdOx1 nCoV-19 vaccine (three patients after the 1st dose and one patient after the 2nd dose) and from 4 to 8 weeks (mean 6.4) after mRNA-1273 vaccine (all patients after the 2nd dose). Out of 24, 14 patients (58.3%) experienced a patchy alopecia areata, while an extensive disease occurred in 10/24 patients (41.7%): six patients with whole scalp involvement (alopecia areata totalis) and four patients with the whole body affected (alopecia areata universalis). Relapse of prior alopecia areata occurred in 22 patients (91.7%), while in the remaining two cases alopecia appeared de novo (8.3%). Twelve patients reported a history of autoimmune disease (50%), particularly Hashimoto thyroiditis (nine patients), celiac disease (two patients) and undifferentiated connective tissue disease (one patient). Treatment of alopecia areata was chosen based on previous therapies, disease extension and the presence of associated comorbid conditions. Ten patients with patchy alopecia areata were treated with topical 0.05% clobetasol solution, that was associated with topical minoxidil in eight cases. Systemic methylprednisolone and topical 0.05% clobetasol solution were concomitantly prescribed in one patient with patchy alopecia areata and in one with alopecia areata totalis. The combination of oral minoxidil, cyclosporin and topical steroid was used in three patients with alopecia areata totalis and three with alopecia areata universalis. Cyclosporin in combination with topical 0.05% clobetasol solution was the treatment of choice for three patients with patchy alopecia areata, two alopecia areata totalis and one alopecia areata universalis. Both topical and systemic treatments were administered for at least 12 weeks, if tolerated (Table 1).

The overall clinical improvement was associated with a reduction in the mean baseline SALT score from 43.4 (5–100) to 36.6 (3.2–97.7) after 12 weeks (Figure 1).

Indeed, hair regrowth was detected in 16 patients (10 patchy alopecia areata, 3 alopecia areata totalis, 3 alopecia areata universalis), whereas worsening was experienced by four patients (four patchy alopecia areata, one alopecia areata totalis) with persistence of active disease (Figure 2 and Figure 3). In the group of non-responder patients (5/21, 23.8%), two patients were treated with topical 0.05% clobetasol solution (patchy alopecia areata), one patient with systemic methylprednisolone and topical 0.05% clobetasol solution (patchy alopecia areata), one patient with cyclosporine and topical 0.05% clobetasol solution (patchy alopecia areata) and one patient with oral minoxidil, cyclosporine and topical 0.05% clobetasol solution (alopecia areata totalis). Three patients (two alopecia areata totalis, one alopecia areata universalis) were lost to follow-up and were not included in the clinical assessment after 12 weeks of treatment.

## 4. Discussion

The diagnosis of vaccine-induced autoimmune disease is based on a case-by-case evaluation, taking into consideration the temporal relation of vaccination and development of alopecia and the exclusion of common etiologies [9]. Our study did not reveal any association or causal relationship between COVID-19 vaccination and alopecia areata. In our center, the percentage of patients with alopecia areata that occurred during the COVID-19 vaccination campaign was similar to the rate seen prior to the COVID-19 pandemic. We found a higher prevalence of women, probably due to an increased susceptibility of females to autoimmune disorders. Furthermore, females mount more robust and rapid immune responses following vaccination as a result of many biological mechanisms driving sex dimorphism in adverse reactions [11]. Thus, hormonal and genetic factors might contribute to this differential immune response to vaccination [11]. In our study, clinical manifestations varied from patchy alopecia in more than half of patients (58.3%) with a greater involvement occurring in 10/24 patients (41.7%); as titer of specific anti-SARS-CoV-2 antibodies was not performed, severity grade could not be related to the magnitude of vaccination-related immune response. Alopecia areata occurred from 1 to 16 weeks after COVID-19 vaccination, demonstrating high variability in the timing of onset, ranging from a few days to 3 weeks in patients who had ChAdOx1 nCoV-19 vaccination, and after 1 to 3 weeks with BNT162b2 vaccine [5,6,7,8,9,10]. Overall, we observed the development of alopecia areata after a mean period of 5.2 weeks, with a shorter mean period of disease onset after ChAdOx1 nCoV-19 vaccination (2.2 weeks), and a longer mean period of disease onset after mRNA-1273 vaccination (6.4 weeks). The mean value was 5.7 weeks after exposure to BNT162b2. The occurrence of different subtypes of alopecia areata has been related to COVID-19 vaccines, including focal well-circumscribed patches of hair loss, but also more severe manifestations involving the whole scalp (alopecia areata totalis) or both scalp and body hair (alopecia areata universalis) [4,5,6,7]. In line with our findings, Scollan et al. described nine cases of alopecia areata, including six patchy alopecia areata and three alopecia areata universalis that developed within 1 week to 16 weeks after BNT162b2 vaccine and within 4 to 8 weeks after mRNA-1273 vaccine [4]. Likewise, a growing number of alopecia areata cases related to COVID-19 vaccination have been reported with time intervals similar than those observed in our study group. Patients receiving ChAdOx1 nCoV-19 vaccine developed clinical manifestations from a few days to a few weeks, increasing up to 16 weeks with BNT162b2 vaccine. After the administration of mRNA-1273 vaccine, the time period was estimated as being 4 to 8 weeks [5,6,7,8,9,10].

Most of our patients (76.2% (16/21)) had an overall improvement of clinical signs, with mean SALT score reduction after 12 weeks of topical and/or systemic therapies. Patients with patchy alopecia areata were mostly treated with topical high-potency corticosteroid (topical 0.05% clobetasol solution), in combination with systemic treatments (corticosteroids, cyclosporin, oral minoxidil) in the case of a more severe hair loss involving the whole scalp or the entire body. Previous observation related to alopecia following COVID-19 vaccination reported different therapeutic approaches, including systemic tofacitinib citrate for patchy alopecia areata and alopecia areata universalis, as well as topical corticosteroids for single lesions on the scalp [4,6]. Topical immunotherapy with squaric acid dibutylester combined with 5% minoxidil was also the therapy of choice for alopecia areata universalis, while triamcinolone alone or in conjunction with topical tacrolimus and 5% minoxidil was used for patchy alopecia areata [8,9,10]. Albeit an appropriate approach is usually prescribed, therapeutically resistant alopecia areata following the second dose of ChAdOx1 nCoV-19 and BNT162b2 vaccine has already been described [12]. The occurrence of autoimmune disorders after COVID-19 vaccines has been reported and the eventual risk of vaccines to favor autoimmunity has raised public concerns, though the causal relationship has not been demonstrated and the pathogenic mechanism underlying the potential development of autoimmune disorders after vaccination remains to be elucidated [13,14,15]. The putative mechanism related to COVID-19 vaccination which might trigger the onset of autoimmune manifestations includes molecular mimicry and the production of autoantibodies. Similarities in aminoacidic sequences between still unknown human self-antigens and SARS-CoV-2 proteins may hypothetically induce immune cross-reactivity against self-antigens [14,15]. Moreover, vaccines may predispose to the development of alopecia areata because of the rapid and potent induction of type I interferon (IFN) expression, boosting antibody production against the virus and the secretion of several cytokines, such as interleukin-12 and interleukin-23. Polymorphisms in the genes encoding such interleukins or their receptors have been associated with a susceptibility for autoimmunity. In addition, an exaggerated production of type I IFNs may contribute to hampering immune tolerance and consequently to autoimmunity [13]. These findings suggest a potential role of vaccines to trigger autoimmunity, as observed in patients vaccinated for hepatitis B and hepatitis A, that showed a significantly higher prevalence of alopecia areata [16]. An increased T cell activation releasing IFN-γ and tumor necrosis factor-α around hair follicles can be determined by some viruses [16,17]. In a recent systematic review, relapse rate of alopecia areata was estimated as 42.5% after SARS-CoV-2, compared with a significantly lower recurrence (12.5%) in patients without SARS-CoV-2 infection, suggesting a role of this virus in worsening alopecia areata in individuals with a pre-existing diagnosis. The boosting effect by SARS-CoV-2 infection on skin immune disorders is suggested by the relapse of common immune-mediated disease described after SARS-CoV-2 infection, whereas less likely new-onset immune diseases have also been described [18,19]. In particular, only 0.2% of the patients belonging to a large cohort of COVID-19-positive individuals (7958 patients) with no history of alopecia areata developed a new diagnosis of alopecia areata after viral infection [20]. Other components involved in adverse reactions to vaccines are represented by vaccine adjuvants. Adjuvanted COVID-19 vaccines might elicit an autoimmune/autoinflammatory syndrome through NOD-like receptor (NLR) pyrin domain containing 3 inflammasome [13,21]. The NLRP3 inflammasome has a key function in innate and adaptive immune systems, possibly contributing to several autoimmune diseases [13,22]. Notwithstanding the limited evidence linking vaccines to autoimmune disorders, social media generated concerns and hesitancy about COVID-19 vaccination, contributing to psychological stress regarding its acceptance. This stressful condition in patients receiving vaccination might eventually induce alopecia areata after the 1st dose. Potential triggers and medical conditions predisposing or associated with the onset of alopecia areata were investigated and ruled out, though the stress related to COVID-19 vaccination itself as potential trigger cannot be excluded. Nevertheless, no evidence supporting this hypothesis could be provided. As most patients included in our case series had previous history of alopecia areata, and half of them reported history of autoimmune disorders, the contribution of vaccines in causing flares, through an immune dysregulation, should be considered, especially in these populations.

Sampling bias is one of the limitations of this study as we included a population referring to the ambulatory of hair disorders. We sought to provide clinically relevant information to manage alopecia areata, even with an extensive involvement, occurring after COVID-19 vaccination, whereas no mechanistic or epidemiological findings were meant to be included.

## 5. Conclusions

This study provided additional data related to hair loss following COVID-19 vaccination. Physicians should be aware about the eventual occurrence of clinical manifestations of alopecia areata in the first 16 weeks after vaccine exposure, especially in patients with a personal history of autoimmune disorders. However, we remarked the overwhelming benefits provided by COVID-19 vaccination; the number of patients with alopecia areata seen during the COVID-19 vaccination campaign was not statistically different from that referring to the same service prior to the COVID-19 pandemic and, thereby, it should not discourage immunization campaigns that were overall safe.

## Figures and Tables

**Figure 1 vaccines-10-01467-f001:**
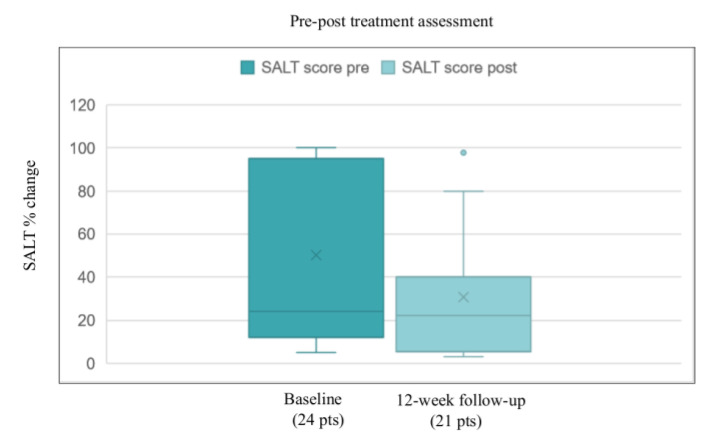
Pre–post-treatment assessment of SALT score: change in the mean value from 43.4 (5–100) to 36.6 (3.2–97.7) after 12 weeks.

**Figure 2 vaccines-10-01467-f002:**
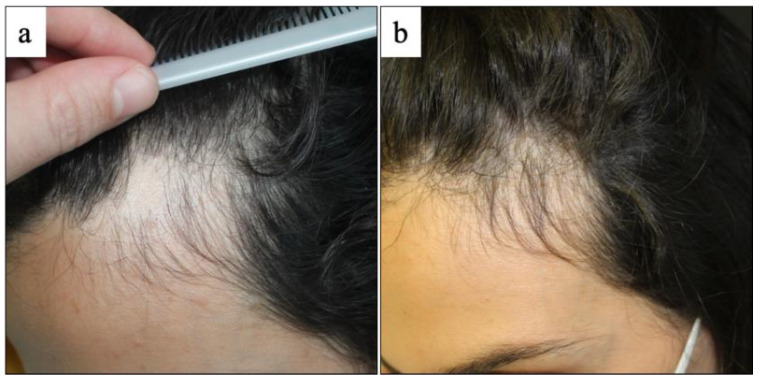
Clinical evaluation of alopecia areata at baseline and at 12 weeks of treatment: (**a**) presence of a well-defined oval patch of alopecia located on the left fronto-parietal region in a 42-year-old female with history of alopecia, exhibiting (**b**) hair regrowth after application of topical 0.05% clobetasol solution.

**Figure 3 vaccines-10-01467-f003:**
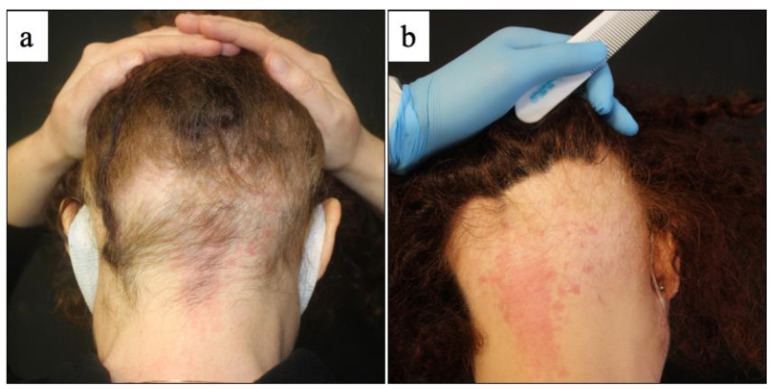
Clinical evaluation of alopecia areata at baseline and at 12 weeks of treatment: (**a**) a large area of alopecia (ophiasis pattern) on the occipital scalp in a 39-year-old female with history of alopecia, showing worsening (**b**) of hair loss after treatment with systemic cyclosporin and topical 0.05% clobetasol solution.

**Table 1 vaccines-10-01467-t001:** Demographic and clinical characteristics of the patients: data related to vaccines and therapy.

Clinical Data: No (%)		Count

**Sample**		24 pts
**Sex**	Male	4 (16.7)
	Female	20 (83.3)
**Age**		Mean value 39.1 (14–66 age)
**COVID-19 vaccine**	BNT162b2	15 (62.5)
	mRNA-1273	5 (20.8)
	ChAdOx1 nCoV-19	4 (16.7)
**Onset of AA after vaccine**	BNT162b2	Mean 5.7 (1 week–16 weeks)
	mRNA-1273	Mean 6.4 (4 weeks–8 weeks)
	ChAdOx1 nCoV-19	Mean 2.2 weeks (1 week–3 weeks)
**Clinical subtype**	Patchy AA	14 (58.3)
	AAT	6 (25)
	AAU	4 (16.7)
**Treatment regimen**	Patchy AA—14 pts (58.3)	Topical steroids—10 pts (41.6) Systemic and topical steroids—1 pt (4.2) Cyclosporin and topical steroids—3 pts (12.5)
	AAT—6 pts (25)	Oral minoxidil, cyclosporin and topical steroids—3 pts (12.5) Cyclosporin and topical steroids—2 pts (8.4) Systemic and topical steroids—1 pt (4.2)
	AAU—4 pts (16.7)	Oral minoxidil, cyclosporin and topical steroids—3 (12.5) Cyclosporin and topical steroids—1 (4.2)

**Comorbidities**	Autoimmune disease—12 pts (50)	Hashimoto thyroiditis—9 pts (37.5) Celiac disease—2 pts (8.3) Undifferentiated connective tissue disease—1 pt (4.2)

AA = alopecia areata; AAT = alopecia areata totalis; AAU = alopecia areata universalis; fu = follow up; pts = patients.

## Data Availability

Not applicable.

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
