# Peer review of "Alopecia Areata Occurring after COVID-19 Vaccination: A Single-Center, Cross-Sectional Study"

_vaccines, 2022, doi:10.3390/vaccines10091467_

Round 1

Reviewer 1 Report

The authors faced a very interesting topic. Adverse reactions after vaccination against Srars-CoV2/Covid19 was (is) a very sensitive subject, given their impact on the public opinion. For this reason I think that the manuscript is not strong enough for the publication. The experimental design is too weak: the sample size (n=24) is too small, there is no comparison to the "control" group (i.e. vaccinated/no alopecia patients) and no statistics are provided

I really think that the topic can be of interest for the scientific community (and not only) but the authors should re-think to the experimental design and analyze data under strict statistical approaches.

In my opinion the manuscript is not acceptable for the publication in Vaccines

Author Response

Response to Reviewer 1 Comments

Point 1: The authors faced a very interesting topic. Adverse reactions after vaccination against Sars-CoV2/Covid19 was (is) a very sensitive subject, given their impact on the public opinion. For this reason I think that the manuscript is not strong enough for the publication. The experimental design is too weak: the sample size (n=24) is too small, there is no comparison to the "control" group (i.e. vaccinated/no alopecia patients) and no statistics are provided

I really think that the topic can be of interest for the scientific community (and not only) but the authors should re-think to the experimental design and analyze data under strict statistical approaches.

Response 1: Dear reviewer, according to your suggestions we defined a control group deriving from the dedicated ambulatory of hair disorders and including patients referring to this service in 1-year period (31st December 2018 – 31st December 2019) during pre-COVID era.

Out of 440 charts of patients referring to the ambulatory of hair disorders during the period 31st March 2021- 31st March 2022, 74 subjects (16.8%) were affected by alopecia areata. This percentage of patients affected by alopecia areata was in line with the rate of patients (14%, 68/484) with a diagnosis of alopecia areata observed in the same service prior to COVID-19 pandemic (31st December 2018 – 31st December 2019). The slight increase of patients’ percentage during COVID-19 was not statistically significant (p=0.244). (Page 2, lines 80-82; lines 88-94; page 3 lines 99-104)

Reviewer 2 Report

In this work, Tassone et al report the onset of alopecia areata following COVID19 vaccination. In the discussion they also provide some possible explanation for this, which I found it interesting.

Despite the work is well described and report both positive and negative reactions to the treatment that patients received, there are some points that it would have been nice to address and take into consideration.

1) The introduction is really minimal and it would be nice to report here if there are already described cases of alopecia worsening of arising after vaccination.

2) Among the patients there are people already suffering of autoimmune disorders, it would have been interesting to evaluate the inflammation status of these patients and evaluate also if the symptoms related to their pathology changed after the vaccination.

3) Do they authors think that there is a correlation between the response to the vaccine and the grade of the alopecia? Meaning: did they ever measure the titer of specific antibodies induce after the vaccination? Is there any correlation between the onset of alopecia and the way the patient reacted to the vaccine? Did the patients experience other secondary effects to the vaccination, more than the common ones reported in the population?

4) Are there cases of Alopecia Areata reported after Sars CoV2 natural infection? If yes, how does this relate to what the authors describe here? Is the incidence high/low? Same percentage?

5) The patients analyzed are mostly women. Is there a bias in the selection of the group or is there a specific reason? Are women more prone to develop the Alopecia Areata? Is this a specific reaction to the vaccine?

If the authors could discuss some of these point in their manuscript would increase the quality of the report.  Since the manuscript belongs to the special issue on safety an autoimmune response to SARS COV2 vaccination, I would also suggest the authors to discuss a bit more this aspect in their manuscript, taking into consideration already reported case of autoimmunity after the vaccination. It would be nice to see how their research fit into the present scenario of adverse effects to the vaccine.

Author Response

Response to Reviewer 2 Comments

Point 1: The introduction is really minimal and it would be nice to report here if there are already described cases of alopecia worsening of arising after vaccination.

Response 1: Data reported in literature have been summarized in introduction (page 2, lines 54-58)

Point 2: Among the patients there are people already suffering of autoimmune disorders, it would have been interesting to evaluate the inflammation status of these patients and evaluate also if the symptoms related to their pathology changed after the vaccination.

Response 2: We do not have further information about the inflammation status of these patients as no additional tests are routinely required.

Point 3: Do they authors think that there is a correlation between the response to the vaccine and the grade of the alopecia? Meaning: did they ever measure the titer of specific antibodies induce after the vaccination? Is there any correlation between the onset of alopecia and the way the patient reacted to the vaccine? Did the patients experience other secondary effects to the vaccination, more than the common ones reported in the population?

Response 3: as titer of specific anti-SARS-CoV-2 antibodies was not performed, severity grade could not be related to the magnitude of vaccination-related immune response. We have included a sentence in discussion (Page 6, lines 192-194).

Patients included in the study denied further adverse reactions to vaccine.

Point 4: Are there cases of Alopecia Areata reported after Sars CoV2 natural infection? If yes, how does this relate to what the authors describe here? Is the incidence high/low? Same percentage?

Response 4: In a recent systematic review relapse rate of alopecia areata was estimated to be 42.5% after SARS-CoV-2, compared with a significant low recurrence (12.5%) in patients without SARS-CoV-2 infection, suggesting a role of this virus in worsening alopecia areata in individuals with a preexisting diagnosis…

We have added more details in discussion (Page 7, lines 245-255)

Point 5: The patients analyzed are mostly women. Is there a bias in the selection of the group or is there a specific reason? Are women more prone to develop the Alopecia Areata? Is this a specific reaction to the vaccine?

Response 5: We found a higher prevalence of women, probably due to an increased susceptibility of females to autoimmune disorders. Furthermore, females mount more robust and rapid immune responses following vaccination as a result of many biological mechanisms driving sex dimorphism in adverse reactions…

We have included some sentences to find an explanation of gender prevalence in discussion (Page 6, lines186-190)

Reviewer 3 Report

Tassone et al reported twenty-four patients, who were examined for the occurrence of alopecia areata within 16 weeks after COVID-19 vaccination. Out of 24, 14 patients experienced a patchy alopecia areata, while an extensive disease occurred in 10/24 patients: 6 patients with whole scalp involvement (alopecia areata totalis) and 4 patients with whole body affected (alopecia areata universalis). After treatment, mean baseline values of Severity of Alopecia Tool (SALT) score decreased from 43.4 to 36.6 after 12 weeks of treatment, with evidence of hair regrowth in 16/21 patients. These findings suggest a correlation between COVID-19 vaccination and alopecia areata. The content is interesting, but I think the tables or figures are inadequate.

major concerns)

1) Patient backgrounds and figures are given in the text, but they are very difficult to understand on their own. Patient background (i.e. age, sex, onset of alopecia areata, comorbidities, previous treatments for alopecia areata and any concomitant medications, vaccine type (BNT162b2, ChAdOx1 nCoV-19, mRNA-1273, etc), type of AA, serum markers for liver, kidney, and thyroid function, etc.)needs to be summarized in a graph. Some graphs of treatment progress, such as SALT, are also needed for better understanding of readers.

2) From the results of this study, it does not seem possible to say that AA was increased (or exacerbated) by the vaccine. Is there a difference between the percentage of people who develop AA after vaccination and the percentage who develop AA in the general population? If there is a difference there, you could say that it was induced or exacerbated by the vaccine.

Author Response

Response to Reviewer 3 Comments

Point 1: Patient backgrounds and figures are given in the text, but they are very difficult to understand on their own. Patient background (i.e. age, sex, onset of alopecia areata, comorbidities, previous treatments for alopecia areata and any concomitant medications, vaccine type (BNT162b2, ChAdOx1 nCoV-19, mRNA-1273, etc), type of AA, serum markers for liver, kidney, and thyroid function, etc.)needs to be summarized in a graph. Some graphs of treatment progress, such as SALT, are also needed for better understanding of readers.

Response 1: A table with patient background (page 4) and a graph exhibiting SALT score changes after 12 weeks (page 5) were added to summarize data provided in the text

Point 2: From the results of this study, it does not seem possible to say that AA was increased (or exacerbated) by the vaccine. Is there a difference between the percentage of people who develop AA after vaccination and the percentage who develop AA in the general population? If there is a difference there, you could say that it was induced or exacerbated by the vaccine.

Response 2: Dear reviewer, according to your suggestions we defined a control group deriving from the dedicated ambulatory of hair disorders and including patients referring to this service in 1-year period (31st December 2018 – 31st December 2019) during pre-COVID era.

Out of 440 charts of patients referring to the ambulatory of hair disorders during the period 31st March 2021- 31st March 2022, 74 subjects (16.8%) were affected by alopecia areata. This percentage of patients affected by alopecia areata was in line with the rate of patients (14%, 68/484) with a diagnosis of alopecia areata observed in the same service prior to COVID-19 pandemic (31st December 2018 – 31st December 2019). The slight increase of patients’ percentage during COVID-19 was not statistically significant (p=0.244). (Page 2 lines 80-82; lines 88-94; page 3 lines 99-104)

Reviewer 4 Report

General comment-This is a clearly presented case study demonstrating the association of COVID-19 vaccination and alopecia areata. Limited reports suggested the potential causal association between COVID-19 vaccines and development/worsening of alopecia areata. Alopecia areata is an autoimmune condition characterized by patches of baldness (scalp/body) with potential evolution of total hair loss. In this manuscript, the authors examined the relationship between COVID-19 vaccines and alopecia areata using retrospective observational study. Twenty four patients were observed for the occurrence of alopecia areata in the first 16-weeks of COVID-19 vaccination.The following study suggested the potential correlation between COVID-19 vaccination and alopecia areata particularly in patients with a personal/family history of autoimmune disorders.

Summary of the salient findings:

Out of 24 patients, 14 experienced patchy alopecia, 10 patients experienced an extensive disease, 6 patients with whole scalp involvement and 4 patients with whole body affected. Fifty percent of the patients have a history of autoimmune disease. Pharmacological treatment of alopecia(using topical/systemic agent) reduced alopecia score after12-wk with evidence of hair regrowth.

The proposed study is very interesting, but I have the following comments and concerns.

1.    This observational study revealed the results of alopecia in 24 subjects with three different COVID 19-vaccines (BNT162b2; 15 cases, 1273 vaccine; 5 cases; ChAdOx1 nCoV-19, 4 cases) and SALT score after treatments. It will be very important to demonstrate these results in tabulated form also.

2.    Though authors have discussed potential causes of autoimmune disease  manifestations in response to COVID-19 vaccination. It will be useful to show/demonstrate these potential mechanisms in a plausible model form.

3.    Minor comment- Line 219-227 (Appendix, Fig1,2)- Looks like Fig. legends are just repeated in these lines, can be removed. As same figure legends are already shown under the respective Figure 1 (128-131)  and Figure 2 (137-140).

I recommend the manuscript be accepted for publication, with addressing these concerns  

Author Response

Response to Reviewer 4 Comments

Point 1: This observational study revealed the results of alopecia in 24 subjects with three different COVID 19-vaccines (BNT162b2; 15 cases, 1273 vaccine; 5 cases; ChAdOx1 nCoV-19, 4 cases) and SALT score after treatments. It will be very important to demonstrate these results in tabulated form also.

Response 1: A table with patient background (page 4) and a graph exhibiting SALT score changes after 12 weeks (page 5) were added to summarize data provided in the text

Point 2: Though authors have discussed potential causes of autoimmune disease  manifestations in response to COVID-19 vaccination. It will be useful to show/demonstrate these potential mechanisms in a plausible model form.

Response 2: We fully agree that potential mechanisms associating COVID-19 vaccination to the occurrence of alopecia areata would add a greater value to our study, but herein we aimed to retrospectively describe cases of alopecia areata occurring after COVID-19 vaccination without providing mechanistic findings that support a causal relationship. Thus, we clarified that this is a time-dependent association only (page 7 lines 264-267, Page 8 lines 274-277)

Point 3: Minor comment- Line 219-227 (Appendix, Fig1,2)- Looks like Fig. legends are just repeated in these lines, can be removed. As same figure legends are already shown under the respective Figure 1 (128-131)  and Figure 2 (137-140).

Response 3: Appendix B with figure legend was removed

Round 2

Reviewer 1 Report

I'm really sorry but I still think that the showed results are not enough for the publication.

This is the point: "The slight increase of patients’ percentage during COVID-19 was not statistically significant (p=0.244)" and, for this reason, the authors cannot state that "increasing data suggest a real association" (in the title too).

The faced topic is too important to draw conclusions based on 24 patients ("Our findings further suggest a correlation between COVID-19 vaccination and alopecia areata"?)

I suggest to the editor to reject the manuscript and to the authors to deeply revise their work and try to better describe their results: may be could be good to change point of view, better supporting the vaccination against Sars-CoV-2 because no evidence of correlation with alopecia was proven.

Author Response

We appreciated your clarifications and suggestions. This study is not aimed to analyze risk or association between COVID-19 vaccination and the onset of alopecia areata, but to simply describe clinical features of alopecia areata occurring after COVID-19 vaccination. This observation is in line with other papers, consisting of small case series or single case reports. Similarly, we are not providing evidence of causal correlation between vaccination and alopecia areata. We agree we cannot draw conclusions and, indeed, we sought to lessen the main message as we have already done in the whole manuscript. Our results suggested on the contrary no increase rate in the occurrence of alopecia during COVID-19 vaccination. Herewith, we reported the latest modifications we performed, tracked in the text:

We modified the title according to your suggestions (“Alopecia areata occurring after COVID-19 vaccination: a single-center, cross-sectional study”)

Line 19 (abstract): “Recently, few reports highlighted the potential causal association between the administration of COVID-19 vaccines and the development, worsening or recurrence of alopecia areata” was replaced with “Recently, few reports described the development, worsening or recurrence of alopecia areata after the administration of COVID-19 vaccines”.

Line 35 (abstract): “Conclusion. Our findings further suggest a correlation between COVID-19 vaccination and alopecia areata, particularly in patients with a personal or family history of autoimmune disorders, though pathogenic mechanisms underlying this observation have not been clarified yet” was replaced with “Conclusion. This study described the occurrence of alopecia areata after COVID-19 vaccination and its management that implicates the use of both topical and systemic therapies”.

Line 45 (introduction): “A few reports highlighted a potential causal association between COVID-19 vaccination and the development, worsening or recurrence of alopecia areata” was replaced with “The development, worsening or recurrence of alopecia areata, an autoimmune condition, has been described after COVID-19 vaccination [4-10]”.

Line 99 (results): “The slight increase of patients’ percentage during COVID-19 was not statistically significant (p=0.244)” was replaced with “The number of cases occurring during COVID-19 vaccination campaign was not statistically different from those referring at the same service prior to COVID-19 pandemic (p=0.244)”.

Line 102 (results): “The analysis of medical records identified 24 of them (32.4%, 24/74) with potential time-related link with COVID-19 vaccines” was replaced with “The analysis of medical records identified 24 of them (32.4%, 24/74) occurring after COVID-19 vaccination.”

Line 181 (discussion): “According to our observations, diagnostic rate of alopecia areata was similar prior COVID-19 and during pandemic period” was replaced with “Our study did not reveal any association or causal relationship between COVID vaccination and alopecia areata. In our center, the percentage of patients with alopecia areata that occurred during COVID vaccination campaign was similar to the rate seen prior to COVID-19 pandemic”.

Line 225 (discussion): “Considering the mounting data supporting a possible link between COVID-19 vaccines and autoimmune disorders, the concrete risk of vaccines to favor autoimmunity has raised public concerns” was replaced with “The occurrence of autoimmune disorders after COVID-19 vaccines has been reported and the eventual risk of vaccines to favor autoimmunity has raised public concerns”

Line 278 (conclusions): “Physicians should consider the potential association between COVID-19 vaccination and clinical manifestations of alopecia areata” was replaced with “Physicians should be aware about the eventual occurrence of clinical manifestations of alopecia areata in the first 16 weeks after vaccine exposure, especially in patients with a personal or family history of autoimmune disorders.”

The importance of vaccines and their distribution is highlighted in conclusions (line 280) “However, we remarked the overwhelming benefits provided by COVID-19 vaccination: the number of patients with alopecia areata seen during COVID-19 vaccination campaign was not statistically different from those referring at the same service prior to COVID-19 pandemic and, thereby, it should not discourage the immunization campaigns that resulted overall safe” and in results (line 99) “The number of cases occurring during COVID-19 vaccination campaign was not statistically different from those referring at the same service prior to COVID-19 pandemic (p=0.244)”

Reviewer 3 Report

No additional comments. The authors have answered appropriately.

Author Response

No additional comments were required by the reviewer.